# IAP-Mediated Protein Ubiquitination in Regulating Cell Signaling

**DOI:** 10.3390/cells9051118

**Published:** 2020-04-30

**Authors:** Baptiste Dumétier, Aymeric Zadoroznyj, Laurence Dubrez

**Affiliations:** 1Institut National de la Santé et de la Recherche Médicale (Inserm), LNC UMR1231, 21000 Dijon, France; Baptiste.Dumetier@u-bourgogne.fr (B.D.); Aymeric.Zadoroznyj@u-bourgogne.fr (A.Z.); 2Université de Bourgogne Franche-Comté, LNC UMR1231, 21000 Dijon, France

**Keywords:** IAP, ubiquitination, cell signaling, inflammation

## Abstract

Over the last decade, the E3-ubiquitine ligases from IAP (Inhibitor of Apoptosis) family have emerged as potent regulators of immune response. In immune cells, they control signaling pathways driving differentiation and inflammation in response to stimulation of tumor necrosis factor receptor (TNFR) family, pattern-recognition receptors (PRRs), and some cytokine receptors. They are able to control the activity, the cellular fate, or the stability of actors of signaling pathways, acting at different levels from components of receptor-associated multiprotein complexes to signaling effectors and transcription factors, as well as cytoskeleton regulators. Much less is known about ubiquitination substrates involved in non-immune signaling pathways. This review aimed to present IAP ubiquitination substrates and the role of IAP-mediated ubiquitination in regulating signaling pathways.

## 1. Introduction

IAPs (Inhibitors of Apoptosis) constitute a family of conserved proteins found in a variety of organisms including virus, yeast, fungus, worms, insects, arthropods, fish, and mammals [1,2,3,4,5,6]. They were discovered in baculovirus in a genetic screen aiming to identify viral proteins able to prevent the death of infected cells [1]. Mammal IAPs share this anti-apoptotic property (for review, see Reference [7]). However, most of them function primarily as cell signaling regulators [8,9]. cIAP1 (cellular IAP1), cIAP2, and XIAP (X-linked IAP) are potent regulators of innate immune response [10]. They control the activation of NF-κB (nuclear factor kappa-light-chain-enhancer of activated B cells) and MAPK (mitogen activated protein (MAP) kinase) signaling pathways in response to the stimulation of members of the tumor necrosis factor receptor (TNFR) superfamily, pattern-recognition receptors (PRRs), or some cytokine receptors. Their signaling activity is directly related to their ability to promote ubiquitination of key signaling intermediates [11]. 

Ubiquitination is a post-translational modification initially described as the principal way to regulate the half-life of proteins and to clear defective and misfolded proteins. The Nobel Prize rewarded the discovery of ubiquitin-mediated protein degradation by Aaron Ciechanover, Avram Hershko, and Irvin Rose in 2004. Since then, the increasing interest of researchers for the ubiquitin system has considerably enlarged the spectral of action of this modification, which has ultimately appeared as a very complex system able not only to control the protein turnover but also to ensure spatiotemporal signal transduction and dynamic communication within the cells. Because they can be implemented very quickly, they constitute a remarkable and effective protein-regulating system employed in the first line of defense by cells to cope with changing cellular environment. They largely contribute to the adaptation of cell behavior to physiological intracellular variations, as well as to intracellular or environmental stresses. 

Out of the eight IAPs found in mammals, cIAP1, cIAP2, XIAP, ML-IAP (Melanoma IAP also named livin), and ILP2 (IAP-like protein 2) are component of the ubiquitination machinery by acting as E3-ubiquitine ligases. They can catalyze the conjugation of K6, K11, K27, K48, and K63-linked ubiquitin chains [12,13,14,15], as well as mono-ubiquitination [16,17,18,19]. They can promote the conjugation of diverse ubiquitin-linked chains on the same substrate leading to variable cellular fate [20]. The conjugation of the ubiquitin-like modifier NEDD8 (neural precursor cell expressed developmentally downregulated protein 8) by IAPs has even been observed [21]. Many IAP-ubiquitination substrates involved in various cellular processes have been reported. Although the role of IAPs in immune response has started to be well characterized, little is known about their function in other signaling pathways (e.g., autophagy, DNA damage response, transcriptional regulation, etc.). This review is an overview of the most documented IAP ubiquitination substrates and the role of IAP-mediated protein modifications in regulating cell signaling. 

## 2. The Ubiquitin System

Typically, ubiquitination consists in the covalent conjugation of ubiquitin moieties on a lysine residue from an intracellular protein substrate. Ubiquitin is a 76 amino-acid peptide that contains seven lysine residues (K6, K11, K27, K29, K33, K48, K63) that, just like the amino-terminal Methionine (M1), behave as acceptor residues for ubiquitin moieties. Thus, intracellular proteins can be modified by a single ubiquitin molecule (monoubiquitination) or multimeric ubiquitin chains (polyubiquitination) of variable topologies conferring specific properties to the modified proteins. Moreover, one protein substrate can be modified on several lysine residues, and the location of the modification is also determinant to modulate the properties of the protein [22]. Ubiquitin moieties are specifically recognized by effector proteins harboring linkage-specific ubiquitin-binding modules that engage in downstream events. For example, proteins modified by conjugation of homotypic K48-linked ubiquitin chains are guided toward the proteasome for degradation, homotypic K63-, or the linear homotypic M1-linked ubiquitin chains are signals for the assembly of intracellular multiprotein signaling platforms, while the mono-ubiquitination of cell-surface receptors triggers their internalization and subsequent lysosomal degradation. Recent observations revealing the presence of mixed chains (composed of multiple linkages), branched chains (in which one ubiquitin unit is modified at two or more acceptor sites), or ubiquitination at non-lysine sites in substrate [23] have added a level of complexity to the ubiquitin code [24]. 

Ubiquitin conjugation is a three-step multi-enzymatic reaction. The first step involves an E1 enzyme that activates, in an ATP-dependent manner, the C-terminus of ubiquitin. Once activated, ubiquitin is transferred to the active cysteine residue of an E2-conjugating enzyme to form an E2-ubiquitin thioester conjugate. E3-ubiquitine ligases promote the ubiquitin transfer from E2s to the substrate [25]. Additional ubiquitins can be tethered to acceptor residues of ubiquitin to form ubiquitin chains. It is generally admitted that E3s support substrate recognition and determine the specific site of the modification, while E2s play a major role in determining the type of ubiquitin chain linkage. However, E3s can also contribute in linkage specificity, particularly in the case of mixed or branched chains, by employing different E2s to elongate the polyubiquitin chain [24,25,26,27]. 

## 3. IAP E3-Ubiquitine Ligases

### 3.1. IAP Structure

IAPs are defined by the presence of one to three specific evolutionary conserved domains named BIR (baculovirus IAP repeat), whose function is protein-protein interaction (Figure 1a,b). cIAP1, cIAP2, and XIAP contain three tandem repeats of BIRs, while ML-IAP and ILP2 display one unique BIR (Figure 1a). These are compact domains composed of approximately 80 amino-acid residues structured in 3 short β-strands and 4-5 α-helices (Figure 1b). In most of them (except the BIR1 of cIAPs and XIAP), the β-strands and the third α-helix form a deep hydrophobic groove that specifically anchors a 4 amino-acid linear motif called IBM (IAP binding motif). The main characteristic of IBMs is the presence of an N-terminal alanine or, for a few of them, a serine residue that has to be exposed to fit into the BIR pocket. IBM has been found in numerous mitochondrial proteins [28], including the second mitochondria-derived activator of caspase (Smac) [29,30], for which the IBM is exposed after removing of the N-terminal mitochondrial targeting motif during its proteolytic maturation process. IBM of non-mitochondrial proteins is located just downstream of the initiating methionine removal by methionine aminopeptidases (in the case of the NF-kappa-B (NF-κB)-inducing kinase (NIK) and the checkpoint kinase 1 (chk1) [31,32] or is exposed at the N-terminal extremity of the cleavage product upon caspase- or calpain-mediating processing during apoptosis or reticular stress respectively (in the case of caspase 3, 7, or 9 or the releasing factor eRF3/GSTP) [33,34,35]. Deciphering the mechanisms of interaction of Smac with the BIRs led to the design of IAP antagonists named Smac mimetics able to bind both the BIR2 and the BIR3 domains (see the reviews in Reference [36,37]). Alternative modes of interaction also exist. The cIAP1-BIR3 can bind protein partners by using unconventional IBMs [38] or another BIR-binding interface [39]. In addition, the BIR1 of cIAP1/2 and XIAP that are unable to anchor IBM bind the signaling adaptor tumor necrosis factor receptor (TNFR)-associated factor 2 (TRAF2) [40,41] and the transforming growth factor-activated kinase 1-binding protein 1 (TAB1) [42], respectively.

The BIRs serve as ubiquitination substrate binding components. Alternatively, substrates can be recruited through adaptors or intermediates, such as TRAF2, TRAF3 [43,44,45], the PHD-bromodomain 1 (TRIP-Br1, Sertad1) [15], or the septin-like ARTS (Apoptosis Related protein in transforming growth factor (TGF)-beta signaling pathway) [46]. 

The E3-ubiquitine ligase activity of cIAP1, cIAP2, XIAP, ML-IAP and ILP2 is supported by the C-terminal RING domain [11] (Figure 1c). RING is the most widespread E3 active domain. It is structured as a zinc finger platform for bringing in close proximity both the ubiquitin-charged E2 and the substrate [47] (Figure 1c). It catalyzes the transfers of ubiquitin moieties from the E2 to a Lysine of the substrate without intermediate conjugation of ubiquitin to E3. 

In addition to the BIRs and RING, cIAPs, XIAP, and ILP2 bear a ubiquitin associated (UBA) domain and cIAPs a caspase activation and recruitment domain (CARD) [11]. The CARD regulates the activating dimerization of cIAPs (see below) [48,49], while the UBA is an ubiquitin binding domain allowing the recruitment of IAPs to signaling complex [50,51]. The UBA is also able to bind ubiquitin-charged E2s suggesting that UBA could contribute in the choice of the E2s, therefore determining the type of ubiquitin chain linkage [12,52]. 

### 3.2. IAP E3-Ubiquitine Ligase Activity

As observed for many RING-finger E3s, a dimerization of IAPs is required for transferring ubiquitin from E2s to the substrate (for review, see Reference [11]). The dimerization interface is located at the extreme C-terminus of the RING (Figure 1c). A crystallographic analysis of a construct encoding the C-part of cIAP1 that includes the BIR3, the RING, and the two intermediate domains, UBA and CARD, revealed that the monomeric cIAP1 adopts a compact inactive closed conformation, in which the RING is sequestered in the heart of the molecule [48]. The monomeric form can bind ubiquitin-charged E2s; however, this binding is not sufficient to promote dimerization [53] nor ubiquitin transfer [54]. The BIR3 and CARD are both critical for maintaining the closed auto-inhibited conformation [49]. Engagement of the BIR3 by synthetic ligands (e.g., Smac mimetics) promotes a cIAP1 conformational modification and RING exposure, resulting in dimerization and subsequent E3-ligase activation, auto-ubiquitination, and degradation [48,54]. A dynamic analysis of cIAP1 conformational change and inter-domain motions suggested that cIAP1 is prime for activation and able to undergo a very rapid activating conformational change upon binding of ligands [54]. When overexpressed in HEK293T cells, cIAPs are unstable proteins because of autoubiquitination and degradation, suggesting their tendency to spontaneously dimerize [55]. Co-expressing the molecular adapter TRAF2 with cIAP1 blocked the cIAP1-auto-ubiquitination process, stabilized the protein [55] and regulated its subcellular localization [56], demonstrating the importance of TRAF2 as regulatory partner of cIAP1. By contrast, the dimeric form of XIAP that does not contain CARD is stable and does not require an activation step [57].

### 3.3. The cIAPs-TRAF2-TRAF3 E3-Ubiquitin Ligase Complex

TRAFs constitute a family of cytoplasmic adaptors involved in signal transduction from a variety of receptors involved in immune response [58]. Just like cIAPs, TRAF1 and 2 were discovery in a biochemical screen aiming to identify cytoplasmic proteins associated with the tumor necrosis factor Receptor 2 (TNFR2) [59]. Since then, 4 other members have been described in mammals, characterized by the structural similarity. They share the presence in protein structure of a specific TRAF domain which function is protein-protein interaction. It binds to receptors by its N-extremity (TRAF-N domain) and to intracellular signaling molecules through its C-extremity (named TRAF-C). The functional form of TRAF proteins is a trimer that binds to trimeric active receptors and connects downstream signaling effectors. Beside this scaffolding function, TRAF2-6 also display an E3-ubiquitin ligase activity thanks to the presence of an N-terminal RING domain [58]. 

Among TRAFs, TRAF2 can directly bind cIAPs, whereas TRAF1 and TRAF3 can bind cIAPs indirectly via TRAF2. The mechanisms of cIAPs–TRAF2 interaction have been extensively analyzed and involved the BIR1 domain. Biochemical and structural studies demonstrated that one cIAP2-BIR1 molecule binds TRAF2 in its trimeric functional form [41]. TRAF2 promotes the recruitment of cIAPs into receptors associated signaling complexes, thereby engaging downstream signaling pathways. The concerned receptors are TNFR1, TNFR2, CD40, CD30, and BAFF-R (B-cell activating factor) from the TNFR superfamily, TLR4 (toll-like receptor-4), nucleotide binding-oligomerization domain (NOD)-like receptors (NLRs), retinoic acid-inducible gene I (RIG-I)-like receptors (RLRs), and some cytokine receptors, such as interferon (IFN) receptor [58,60]. Independently of receptor engagement, cIAP1 has been found associated with TRAF2 in unstimulated cells from several linages, including epithelial cells, lymphocytes, macrophages, and murine embryonic fibroblasts (MEF) [43,61,62].

Increasing evidences suggest that cIAP1 and TRAF2 form an E3-ubiquitin ligase complex, in which cIAP1 functions as E3-enzyme, while TRAF2 constitutes an important regulatory subunit. TRAF2 serves as adaptor for bringing cIAP1 to close proximity of the substrates, it can stabilize cIAP1 by blocking auto-ubiquitination process [55], regulates its subcellular localization [56], and also activates its E3-ubiquitine ligase activity through K63-linked ubiquitination [63]. In some situation, TRAF3 takes part in the complex serving as substrate binding component (Table 1). 

## 4. IAP-Mediated Ubiquitination of Caspases and IAP Antagonists and Regulation of Apoptosis

IAPs were first identified in baculovirus by their ability to block apoptosis of infected cells. In insect cells, they are essential for maintaining cell survival. Downregulation or loss of function of the drosophila IAP1 (DIAP1) is sufficient to induce caspase activation and massive apoptotic cell death in early embryo [109,110]. Mutation within the RING domain abolished the ability of DIAP1 to prevent apoptosis, demonstrating the importance of the E3-ubiquitine ligase activity [109,110]. DIAP1 protein exerts its anti-apoptotic function by inhibiting caspases. It can physically interact with both the initiator and effector caspases and promotes their ubiquitination that results in modulating their stability and activity [111,112]. Apoptosis occurrence requires the neutralization of DIAP1 [112]. Genetic analysis of Drosophila mutants displaying default in developmental cell death highlighted the importance of IAP antagonists [111,112]. These proteins share an N-terminal IBM that competes with the IBM of caspases for DIAP1 interaction. Binding of IAP antagonists to DIAP1 triggers DIAP1 auto-ubiquitination and degradation [112].

In mammals, XIAP and cIAPs are also able to bind and promote the ubiquitination of apoptotic caspases in vitro [16,21,68]. However, the in vivo relevance of this modification is still debated. XIAP is considered as the main direct inhibitor of caspases activity in mammals [113]. It is present in the apoptosome complex, and its binding to initiator caspase-9 is sufficient to prevents the activating dimerization and protease activity of the caspase [114,115]. XIAP can also directly bind and inhibit effector caspases-3 and -7 by hindering the substrate-binding pocket and blocking substrate accessibility [33,34,115,116,117,118,119], independently of ubiquitination process. However, the analysis of thymocytes, embryonic stem cells and fibroblasts from Δ RING mutant-bearing transgenic mice showed an increased caspase activity and an enhanced sensitivity to apoptosis suggesting the physiological requirement of the XIAP E3-ubiquitine ligase activity for caspase-inhibition [118]. Although less critical for apoptosis induction than in drosophila, IBM-containing-IAP antagonists also play a significant role in apoptosis regulation in mammals. The best characterized is Smac that can bind to both the BIR2 and BIR3 of cIAP1, cIAP2, XIAP, and ML-IAP in an IBM-dependent manner [120,121,122]. It is generally admitted that Smac competes with caspases for XIAP interaction, whereby abrogates caspase inhibition, while cIAPs and ML-IAP antagonize Smac-XIAP binding [113,123,124,125]. Moreover, Smac binding to the BIR domains stimulates cIAP1 E3-ubiquitin ligase activity, autoubiquitination, and degradation [126]. Accordingly, Smac deficiency in mice led to increased level of cIAPs demonstrating the essential role for Smac in controlling the intracellular cIAPs content [127]. In turn, ML-IAP, cIAP1, and XIAP can induce the ubiquitination and degradation of their antagonists [73,128,129].

Besides caspases and IAP antagonists, IAPs can also regulate through ubiquination other regulators of apoptosis (Table 1). For most of them, except for the B cell lymphoma 2 (Bcl-2) [46], IAP-mediated ubiquitination results in inhibiting cell death activity. 

## 5. IAP-Mediated Regulation of NF-κB-Signaling

NF-κBs correspond to a family of transcription factors responsible for the expression of several anti-apoptotic genes, cytokines and pro-inflammatory molecules that orchestrate the innate immunity [130]. They are activated upon stimulation of membrane or intracellular receptors, including members of the TNFR superfamily, growth factor receptors, cytokine receptors, and PRRs, as well as in response to environmental or endogenous stress. Two distinct NF-κB activation signaling pathways have been described: the canonical and the non-canonical (Figure 2 and Figure 3). Schematically, the canonical pathway involved the IKK (Inhibitor of κB kinase) complex composed of IKKα, IKKβ, and IKKγ (Figure 2). It is activated by phosphorylation by upstream kinases including the tumor growth factor-β-activated kinase 1 (TAK1), in turn stimulating the UPS (ubiquitin-proteasome system)-mediated degradation of the inhibitor-κB (I-κB) through phosphorylation. This results in the release and translocation of the NF-κB subunits to the nucleus where they promote a specific gene transcription program. The non-canonical pathway is independent of IKKβ and IKKγ (Figure 3). It involves NIK that catalyzes the phosphorylation of IKKα homodimer leading to phosphorylation, processing, and nuclear translocation of the transcription factor [130]. 

### 5.1. XIAP-Mediated TAK1 Ubiquitination

TAK1 is a mitogen-activated protein kinase (MAP3K). Besides its function in transducing MAPK signaling, it can participate to NF-κB activation by phosphorylating IKKβ. It is active when associated to TAK1-binding protein-1 (TAB1) and either TAB2 or TAB3 which share the ability of binding K63-linked ubiquitin chains [131]. Thus, TAK1/TABs complex is recruited to cell signaling platform via binding to K63-linked polyubiquitin scaffold that maintains TAK1 and its substrate in close proximity. TAK1 activation requires a direct K63-linked polyubiquitination. The E3-ubiquitin ligase TRAF6 and XIAP [97,131] can mediate this modification [97,131]. XIAP can directly bind to TAB1 via its BIR1 domain [42,132] and induces K63-linked polyubiquitination that results in TAK1 activation and stabilization of IKK complex [89,97]. The XIAP-mediated TAK1 polyubiquitination is critical for NF-κB activation in response to genotoxic stress [89] and for TGF-β (transforming growth factor beta) signaling [97,133]. XIAP is able to directly interact with the TGF-β receptor type I [97,133] and BMP (bone morphogenetic proteins) receptors [132], bridging the receptor to TAK-1/TABs complex and allowing the engagement of JNK (c-Jun N-terminal kinase) and NF-κB signaling pathways [133]. XIAP has also been involved in a feed-back loop regulation in TGF-β signaling. Indeed, TGF-β can induce the expression of the XIAP gene [96,134]. In turn, XIAP mediates the polyubiquitination and degradation of TAK-1, leading to an inhibition of TGF-β-induced JNK activation [96].

### 5.2. cIAPs-Mediated Ubiquitination of IKKγ and IKKε

IKKγ (also known as NEMO for NF-κB essential modulator) is the regulatory subunit of IKK complex. While the catalytic subunits IKKα and β are activated by phosphorylation, IKKγ is subjected to multiple post-translational modifications including monoubiquitination, K6-linked, K63-linked, M1-linked, and K63/M1 hybrid polyubiquitination [130]. Genotoxic stress or TNFα induce mono- or K6-linked polyubiquitination of IKKγ, respectively, which are inhibited in cIAP1-deficient cells [18,89]. Directed mutagenesis and in vitro assay demonstrated a direct interaction of the N-terminus of IKKγ with the cIAP1 BIR2-3 domain [89] and the ability of cIAP1 to conjugate monoubiquitin at Lys285 [18] and polyubiquitin chains at Lys277 and 309 [89] (Figure 3). Of interest, the *mulluscum contagiosum* virus (MCV) that infects keratinocytes causing small neoplasms takes advantage of this process. The MCV MC159 protein can compete with cIAP1 for IKKγ binding; therefore, it can inhibit K63-linked ubiquitination of IKKγ and can suppress NF-κB activation [135]. 

cIAP1/2 in complex with TRAF2 can also mediate K63-linked ubiquitination of IKKε (also called IKBKE: inhibitor of nuclear factor κ-B kinase subunit ε) [44]. IKKε is a noncanonical member of the IKK family, a downstream effector of TLR3, RIG-1, and IFN-β receptors. It participates in signal transduction leading to the activation of NF-κBs, IFRs (interferon (IFN) regulatory factors) or STATs (Signal Transducers and Activators of Transcription). The K63-linked ubiquitination at K30 and K401 is essential for its kinase activity and NF-κB activation [136].

### 5.3. cIAP1/2-TRAF2-TRAF3 E3-Ubiquitin Ligase Complex Regulates the Cellular Content of c-Rel 

Among NF-κB transcription factors, the c-Rel subunit is required for TLR-induced expression of pro-inflammatory cytokines. It has been associated with inflammatory and autoimmune diseases in c-Rel knockout mouse models. The steady-level of c-Rel and its activation in response to TLR stimulation is enhanced in TRAF2-deficient myeloid cells. TRAF2 mediates UPS-dependent degradation of c-Rel, which depends on the presence of cIAP1s. cIAPs and TRAF2 can complex with c-Rel only in the presence of TRAF3. Thus, cIAPs, TRAF2, and TRAF3 cooperate to regulate the stability of c-Rel [43]. 

### 5.4. Regulation of the Cellular Content of NIK and the Non-Canonical NF-κB-Activating Signaling Pathway by the cIAP1/2-TRAF2-TRAF3 E3-Ubiquitin Ligase Complex

The development of Smac mimetics that trigger cIAP1 auto-ubiquitination and degradation has revealed the role of this IAP in the regulation of the non-canonical NF-κB-activating signaling pathway [137,138]. In the standing state, the cellular content of NIK is maintained low through sustained UPS-mediated degradation process (Figure 3). Cell exposure to Smac mimetics induced NIK stabilization that resulted in the activation of the non-canonical NF-kB signaling [137,138]. 

Genetics analysis of primary multiple myelomas that are characterized by a high level of NIK have revealed inactivating mutations in cIAP-encoding genes [139,140], which strengthen the role of cIAPs in the negative regulation of NIK. Although NIK protein contains an N-terminal IBM that can directly bind the BIR2 cIAPs, TRAF2, and TRAF3 are required for regulating NIK protein turnover [63]. The analysis of TRAFs or cIAPs mutant multiple myeloma and cIAPs- or TRAFs-deficient MEFs demonstrated that NIK degradation is ensured by the TRAF3-TRAF2-cIAP1 ubiquitin ligase complex, in which TRAF3 serves as NIK-binding component and recruits cIAPs via TRAF2 [63,141]. The NIK IBM-cIAPs interaction stabilizes the complex and facilitates the cIAP-mediated NIK degradative (K48-linked chains) ubiquitination [31]. 

## 6. IAP-Mediated Ubiquitination of Receptor-Interacting Kinases (RIPKs)

During the last decade, serine/threonine kinases from receptor-interacting kinase (RIPK) family had emerged as critical determinants of cell fate in response to stimulation of death, interleukin, or pattern-recognition receptors, as well as genotoxic or oxidative stresses, at the crosstalk between differentiation, inflammatory response, and cell death signaling pathways (for review, see Reference [142]). RIPKs are characterized by the presence of a homologous serine-threonine kinase domain (KD) and at least one additional variable domain required for the recruitment of RIPKs into receptor complexes or signaling platforms through homotypic interaction. Their cellular functions are tightly regulated by post-translational modifications, and ubiquitination constitutes one of the most important mechanism regulating their kinase activity, determining their recruitment into various multiprotein signaling complexes and modulating their ability to engage downstream signaling pathways [143,144,145]. cIAP1/2 and XIAP are able to catalyze the conjugation of ubiquitin chains of variable topology to RIPK1, 2, 3, and 4, but the in vivo significance of these modifications is not completely solved [14].

### 6.1. cIAP1/2-Mediated RIPK1 Ubiquitination in Signal Transduction and Ripoptosome Assembly

RIPK1 is a death domain (DD)-containing protein able to bind members of TNFR superfamily and adapter proteins via DD homotypic interaction. It determines the response of cells to receptor stimulation, controlling the activation of transcriptional response leading to survival, differentiation, and inflammation, as well as the assembly of cell death signaling platforms leading to apoptosis or necroptosis. RIPK1 can also take part to TNFR-independent signaling complexes thanks to the presence of a RIP homotypic interaction motif (RHIM) that mediates homotypic interaction with its closely related kinase RIPK3, TRIF (toll–interleukin 1 receptor domain–containing adaptor inducing IFN-β) that is an adaptor downstream of the pathogen-recognition receptors TLR3 and TLR4 [146] and DAI (DNA-dependent activator of interferon-regulatory factor) involved in the RIG-1 (retinoic acid-inducible gene I) signaling pathway. 

The role and mechanisms of regulation of RIPK1 has been extensively studied in TNFR1 signaling pathway and recently very well reviewed [142,147] (Figure 3). Briefly, TNFR1 engagement triggers the binding of RIPK1 along with the adaptor TNFR1-associated death domain protein (TRADD) that allows the recruitment of TRAF2 and cIAP1/2 and the subsequent binding of the linear ubiquitin chain assembly complex (LUBAC). This transient membrane-associated complex referred to complex-I leads to the activation of canonical NF-κB and MAPK that culminate in the expression of genes promoting survival, inflammation, and differentiation. TNFR1 stimulation can also trigger the assembly of RIPK1-containing secondary cytoplasmic complexes referred to as complexes II that ultimately result in apoptotic or necroptotic cell death (Figure 3). While the kinase activity of RIPK1 is required for complexes-II assembly, it is dispensable for its scaffolding function. The ubiquitination of RIPK1 has been shown to be determinant for TNF-α-induced NF-κB activation in some cell lines, such as Jurkat T-cells [20,69,144]. K11 and K63-linked polyubiquitin chains conjugated on the K377 residue of RIPK1 are recognized by the UBD (ubiquitin-binding domain) of the LUBAC subunit HOIP (HOIL1 (interacting protein heme-oxidized IRP2 ubiquitin ligase 1)-interacting protein) and allows the recruitment of LUBAC. In turn, LUBAC added a linear (M1-linked) ubiquitin chain to complex I components. Altogether, K11, K63, M1-linked, and hybrid poly-ubiquitin chains [13,148,149] form a molecular scaffold that allows the recruitment of the trimeric kinase complexes IKK (Inhibitor of κB kinase composed of IKK1/2/NEMO) and TAB1/TAB2/TAK1 and ultimately results in NF-κB and MAPK activation and gene activation (Figure 2). 

cIAP1/2 and XIAP has been shown to promote K11-, K63-, and K43-linked ubiquitination of in vitro RIPK1 and in TNF-α-treated cells [13,14,90,149,150]. We cannot rule out that cIAP1/2 could also mediate the conjugation of branched ubiquitin chains. Of interest, constitutive RIPK1 ubiquitination has been detected in several tumor samples including mucosa-associated lymphoid tissue lymphoma that displays a recurrent t(11;18)(q21; q21) translocation involving cIAP-encoding gene [151]. Exposing cells to Smac mimetics that trigger the degradation of cIAP1 or deletion of cIAPs abrogated TNF-α-dependent ubiquitination of RIPK1 prevented the recruitment of LUBAC, TAK1, and IKK into TNFR1-associated complex I and reduced NF-κB activation [13,149,152]. Although the involvement of RIPK1 in TNFR2 signaling pathway has not been clearly demonstrated, Smac mimetics can also prevent the K63-polyubiquitination of components of TNFR2 signaling complex and inhibit the subsequent recruitment of LUBAC and activation of canonical NF-κB [153]. On the other hand, double deletion of genes *birc2* and *birc3* (encoding cIAP1and cIAP2 respectively) or *birc2* and *birc4* (encoding cIAP1 and XIAP respectively) in mice led to embryogenic mortality, which is abolished or delayed by deleting RIPK1 suggesting that cIAP1 can also inhibit the pro-cell death function of RIPK1 [154]. Deletion of cIAP1/2 sensitized cells to TNF-α, TRAIL (TNF related apoptosis inducing ligand), or Fas-mediated cell death [137,138,155,156,157,158] by promoting the assembly of cytoplasmic RIPK1-containing complex-II [152,159]. Such a cytoplasmic complex, named Ripoptosome, can also be spontaneously formed in cells treated with IAP antagonists or can be assembled in response to microbial infection, genotoxic, or oxidative stress [69,159,160]. In addition to its capacity to assist the scaffolding function of RIPK1, cIAP1 represses its kinase activity and autoactivation required for the assembly of complex II or Ripoptosome [12,161]. The generation of conditional knockin mouse deleted for cIAP2 and expressing cIAP1 mutated in its UBA domain made it possible to distinguish between the cIAP1-mediated RIPK1 ubiquitination required for its scaffolding function and that controlling its kinase activity [12]. These animals developed normally but were highly sensitive to TNF-mediated cell death. UBA mutation did not affect TNF-mediated NF-κB activation but prevented the K48-linked polyubiquitination on several acceptor Lysines of RIPK1 that resulted in sustain activation and accumulation of RIPK1 and enhanced formation of complex-II leading to cell death [12]. Thus, the cIAP1-UBA seems critical in defining the type of ubiquitination and the target lysine residues. This function could be related to its ability to facilitate the recruitment of the E2-ubiquitin conjugating enzyme [52].

To summarize, cIAP1 has the ability to control the scaffolding function of RIPK1 through K11- and K63-linked ubiquitination at K377, as well as its stability and kinase activity required for the cell death-inducing complex II/Ripoptosome assembly, through K48-linked ubiquitination on several ubiquitin-acceptor lysine residues (Figure 2). It constitutes a remarkable flexible checkpoint, able to determine the response to receptor engagement and to regulate the duration of the stimulation by switching the transcriptional response to cell death response or by promoting the ubiquitin-mediated degradation of RIPK1.

### 6.2. Regulation of RIPK3 and Necrosome by IAPs 

In death-inducing complex II or Ripoptosome, RIPK1 binds to its closely related kinase RIPK3 through homotypic interaction via the RHIM domain, forming the so-called necrosome complex (Figure 3). Upon toll-like receptor priming by pathogen-associated molecular patterns (PAMPs), the necrosome can be formed in a cIAP-regulated RIPK1-dependent manner or independently of RIPK1 via a direct binding of RIPK3 with the adaptor TRIF [162]. Depending on the presence or not of adaptors and regulatory proteins, such as XIAP, caspase-8, cFLIPs (Cellular FADD-like IL-1β-converting enzyme-inhibitory protein), or MLKL (Mixed lineage kinase domain like pseudokinase), necrosome can lead to apoptosis, necroptosis and even to activation of inflammasome [162,163,164,165,166,167], leading to interleukin-1 (IL-1) and -18 secretion. Both RIPK3-dependent necrosis and inflammasome activation are negatively regulated by XIAP [163,167,168]. Accordingly, exacerbated pathogen-associated hyper-inflammation has been observed in some patients displaying XIAP deficiency. Moreover, mice lacking cIAP1, cIAP2, and XIAP are predisposed to IL-1-dependent autoantibody-mediated arthritis [166]. Polyubiquitination, including K63-, K48-, and linear-ubiquitination of both RIPK1 and RIPK3, has been detected within the necrosome [20,166]. The exact mechanism and function of this ubiquitination is not clearly determined. It could maintain RIPK1 kinase activity and modulate the assembly of necrosome. Although cIAP1/2 and XIAP are able to bind and catalyze the ubiquitination of in vitro RIPK1 and RIPK3 [14], they have not been involved in such ubiquitination processes [20]. On contrarily, deletion of XIAP in bone marrow-derived dendritic cells resulted in enhanced lipopolysaccharide- and TNFα-mediated necroptosis and inflammation [166]. Deletion of the XIAP-RING domain was sufficient to favor necrosome assembly, suggesting that XIAP functions as E3-ubiquitin ligase in regulating necroptosis. However, the ubiquitination substrates have not been determined [168]. 

### 6.3. XIAP-Mediated RIPK2 Ubiquitination and Regulation of NOD1/2 Signaling Pathway

RIPK2 protein (also named RIP2, RICK, or CARDIAK) lacks DD and RHIM domains, but it contains a CARD allowing its recruitment to the cytosolic receptors NOD1 and NOD2. NOD receptors participate to the innate immune response by sensing intracellular PAMPs. They recognize bacterial peptidoglycans and, in turn, activate pro-inflammatory and antimicrobial response via NF-κB, ERK2, and JNK intracellular signaling pathways. Similarly, to RIPK1, the polyubiquitination of RIPK2 is critical for the recruitment and the activation of TAK1/TAB1/TAB2 and IKK complexes that drive NF-κB and MAPKs signaling pathways [169,170]. Among the IAP family, cIAP1/2 and XIAP are able to induce ubiquitination of RIPK2 [14]. However, XIAP seems the main determinant of NOD signal transduction. NOD2 signaling occurred normally in bone marrow-derived macrophages from cIAP1/2-compromised mice [169,171], while NOD2-dependent activation of NF-κB and p38MAPK and secretion of proinflammatory cytokines were impaired in XIAP-deficient myeloid cells [169,171,172,173]. Accordingly, mice deficient in XIAP are highly sensitive to bacteria [174]. A default in NOD1/2-dependent immune signaling had been detected in XLP2 (X-linked lymphoproliferative syndrome type-2) that is an immunodeficiency disease linked to inactivating mutation in the XIAP gene [169]. XIAP can directly interact with RIPK2 via its BIR2 domain and catalyzes the conjugation of K63-linked ubiquitin chains on K209, K410, and K538 residues located in the kinase domain [173,175,176]. As for RIPK1 in the TNF-signaling pathway, IAP-mediated ubiquitination of RIPK2 allows the recruitment of LUBAC, TAB1/TAB2/TAK1, and IKK complexes and ultimately triggers the activation of transcriptional response [175] (Figure 3). Interfering with XIAP-RIPK2 interaction blocked NOD2-mediated RIPK2 ubiquitination and the downstream signaling pathway [175]. A recent study demonstrated that, similarly to RIPK1 in the TNF-signaling pathway, RIPK2 can form secondary cytosolic high molecular weight complexes [176] upon NOD stimulation. These secondary RIPK2-containing complexes depend on RIPK2 kinase activity and are inhibited by XIAP-mediated ubiquitination [176]. 

## 7. cIAP1-Mediated Degradation of TRAF2/3 as Regulatory Mechanism of Downstream Signaling Pathways 

Because TRAF2 is a critical intermediate in signal transduction from receptors, and TRAF3 serves as substrate binding component in the cIAPs/TRAFs E3-ubiquitin ligase complex, the modulation of TRAF2, and/or TRAF3 cellular content constitutes a very efficient mechanism of spatio-temporal regulation of signaling. 

The maturation and survival of B cells are controlled by CD40 or BAFF-R-dependent activation of the non-canonical NF-κB signaling pathway. Deletion of both cIAP1 and cIAP2 in mice induced a sustained activation of non-canonical NF-κB and maintained B-cells survival and maturation independently of BAFF-R [177]. CD40 or BAFF-R stimulation leads to the recruitment of TRAF3-TRAF2-cIAP1/2 complex to the receptor. In the membrane-associated complex, TRAF2 promotes the K63-linked ubiquitination and activation of cIAPs. In turn, cIAPs induce K48-linked ubiquitination and degradation of TRAF3 [63] and, to a lesser extent, TRAF2 [100] that results in NIK release and engagement of canonical NF-kB signaling [63] (Figure 3). Other members of TNFR can use such mechanisms for the activation of the noncanonical NF-κB pathway [178]. Upon CD40 stimulation, cIAPs and TRAF2 are translocated from the membrane receptor complex into the cytosol in a cIAP-dependent manner. The cytosolic cIAP1/TRAF2 containing-complex promotes MAPK activation [100]. 

A cytoplasmic translocation and cIAP-mediated TRAF2 degradation have also been observed upon TNFR2 [179] or FN14 (Fibroblast growth factor-inducible 14) [180] stimulation. We demonstrated that differentiation of monocytic cells into macrophages is associated with cIAP1-induced UPS-mediated degradation of TRAF2 [62]. However, the significance of this event is still unclear. It could induce the stabilization of NIK and activation of noncanonical NF-kB and/or could constitute a feedback regulatory mechanism inhibiting canonical NF-κB activation [62,181,182,183]. Indeed, the activation of NF-κB in TNFR2-stimulated cells or along the macrophagic differentiation process is transitory, and its downregulation is critical for differentiation achievement. The NF-κB downregulation has been proposed to result from the cIAP1-mediated TRAF2 degradation [62].

The cIAP1-TRAF2-TRAF3 complex is also a potent regulator of TRLs signaling in macrophages [43,101,184,185]. TRAF1 and TRAF3 can directly bind the TRL-associated adaptor MyD88 (Myeloid differentiation primary response 88). They then recruit TRAF2-cIAP1/2 into the receptor-associated complex (Figure 3). RIPK1 and RIPK3 are also present in the complex, recruited via the adaptor TRIF (Figure 3). Depletion of cIAPs by Smac mimetics inhibited TLR2 and TLR4-induced TRAF3-degradation and MAPK signaling but did not affect noncanonical NF-κB [101]. As observed after CD40 stimulation [100], TLR4-associated cIAP-mediated TRAF3-degradation is followed by the cytoplasmic translocation of a secondary large cytoplasmic complex containing, among other signaling molecules, MyD88, TRAF2, and cIAP1/2, which results in canonical NF-κB and MAPK activation [101,185]. In the absence of cIAPs, an XIAP-regulated cytoplasmic Ripoptosome or necrosome can be formed, leading to inflammation and necroptosis [166,167]. 

## 8. IAP-Mediated Regulation of Transcription Factors 

cIAP1 and XIAP indirectly stimulate the activation of the transcription factor Myc and CTF (T-cell factor)/LEF (lymphoid enhancer factor-1) by blocking the recruitment of their respective repressor co-factors Mad1 (Max-dimerization protein-1) and Groucho [17,108]. While cIAP1 promotes the UPS-mediated degradation of Mad1 [108], XIAP catalyzes the mono-ubiquitination of Gro/TLE resulting in a decrease affinity for the transcription factor CTF/Lef. As a consequence, XIAP favors the β-catenin-TCF/Lef complex assembly and the initiation of a Wnt-specific transcriptional program [17]. 

IAPs have also been involved in the regulation of the transcription factor stability. cIAP1 can catalyze the degradative ubiquitination of C/EBP homologous protein CHOP, a transcription factor activated in response to endoplasmic reticulum stress. Consequently, cIAP1 protects pancreatic β-cells from toxic effect of free fatty acids [103]. As previously described, cIAP/TRAFs complex controls the cellular content of c-Rel [43]. In the same way, cIAP/TRAFs complex can mediate the degradation of IFR5 [43] and CREB (C-AMP Response Element-binding protein) [45]. IFR5 and CREB, as well as c-Rel, are mediators of the immune and inflammatory response. IRFs control the expression of IFN and some pro-inflammatory cytokine genes, and CREB plays a specific function in immune response by interfering with NF-κB and IFN response, promoting survival signals in macrophages and controlling the proliferation, survival, and/or differentiation of T and B lymphocytes [186]. IRF5 and CREB can directly interact with TRAF3, which recruits TRAF2 and cIAP to promote ubiquitination and degradation of the transcription factors [43,45].

A positive regulation of the transcriptional activity of IRF1, HIF1α (Hypoxia-inducible factor-1α), and E2F1 (E2 promoter binding factor 1) by IAP-dependent ubiquitination processes has also been reported [104,105,106,187,188]. The activating process of IRFs involves post-transcriptional modifications, such as phosphorylation and K63-linked ubiquitination, which engage their dimerization and nuclear translocation. Molecular adaptors from TRAF family, mainly TRAF6, act as E3-ubiquitin ligase mediating this modification. In 2014, Harikumar et al. demonstrated that cIAP2 in complex with the bioactive sphingolipid mediator sphingosine-1-phosphate (S1P) can bind IRF1 and promotes K63-linked ubiquitination. In astrocytes, cIAP2-mediated IRF1 ubiquitination is essential for IL-1-induced IRF1-dependent expression of the chemokines CCL5 (C-C Motif Chemokine Ligand 5) and CXCL10 (C-X-C motif chemokine 10), which are involved in the recruitment of mononuclear cells to the site of inflammation [187]. HIF1α participates to the adaptive response of cells exposed to stressful conditions. K63-linked ubiquitination is required for HIF1α nuclear translocation and for its recruitment to target promoters. In 2017, Park et al. demonstrated that XIAP can promote this modification and is essential for the expression of HIF-dependent genes in hypoxia-stressed cells [106]. E2F1, the funding member of E2F family, is activated at the end of the G1 phase of cell cycle and promotes the expression of genes required for DNA replication and G1-S cell cycle phase transition. E2F1 also participates to the DNA damage response by controlling cell cycle arrest and apoptosis. The activity and turnover of E2F1 are regulated through post-translational modifications and association with co-factors and regulatory proteins [188]. The accumulation of E2F1 in S phase of cell cycle and in response to DNA damaging agents is associated with K63-linked ubiquitination at lysine 161/164 residues [105]. Studies from our group and others demonstrated the ability of IAPs to bind and regulate the transcriptional activity of E2F1 [104,105,189,190]. We showed that cIAP1 overexpression stabilized E2F1 protein expression and modified its ubiquitination profile, increasing the content of E2F1 modified with K11-and K63-linked polyubiquitin chains and decreasing the proportion of K48-ubiquitinated E2F1 [104]. On the other hand, the downregulation of cIAP1 inhibited the accumulation of E2F1 in S-phase-of cell cycle, as well as the one induced by genotoxic stress [105], and completely blocked its capacity to bind chromatin [104]. cIAP1 can bind E2F1 through its BIR3 domain [189] and can promote K63-linked ubiquitination [105]. Thus, IAPs-mediated K63-linked ubiquitination can act as a signal for recruiting E2F1, as well as HIF1α, to target promoter genes [104,105,106,188].

## 9. IAP-Mediated Regulation of Rho-GTPases and Cytoskeleton Remodeling 

Although sometimes contradictory, numerous studies have reported modifications in the cell shape, cell polarity, or alterations in migratory or invasive behavior of cells after modulation (overexpression or downregulation) of IAP expression in mammal, drosophila, or zebrafish models (for review, see Reference [9]). The ability of IAPs to directly bind members of Rho family support these observations and may explain the divergent results [61,80,82,191]. 

Rho GTPases are potent regulators of actin cytoskeleton that maintain cell architecture and polarization. They orchestrate the actin remodeling involved in several fundamental cellular processes, including cell adhesion, cell motility, cell migration, cell growth, cell differentiation, and cell death. Rho-GTPases constantly cycle between a cytoplasmic, inactive GDP-bound state, and an active GTP-bound state that is mainly found in membrane-associated compartments. While the inactive forms are stabilized by association with RohGDIs (RhoGDP dissociation inhibitors), in their active forms, Rho-GTPases are very unstable proteins. They are either recycled in inactive state by action of the GTPases accelerating proteins (GAPs) or ubiquitinated and degraded by the proteasome system. A function for IAP E3-ubiquitin ligases in the regulation of RhoGTPase turnover has been demonstrated [78,80]. XIAP can promote degradative ubiquitination of Rac1 at Lys147 [80] and cdc42 at Lys166 [78] and cIAP1 can induce ubiquitination of Rac1. Accordingly, loss of XIAP increased the cellular content of Rac1 and cdc42 in normal and tumor cell lines, therefore modifying cell morphology and migration [78,80]. On the other hand, IAP antagonists or silencing of XIAP reduced RhoA activation in response to protease-activated receptor [82], deletion of cIAP1 abolished cdc42 activation, and filopodia formation in response to TNF or EGF(Epidermal Growth Factor) [61] and the overexpression of cIAP2 led to the upregulation and activation of Rac1 in regenerating intestinal epithelial cells [81]. All studies agree that IAPs can bind RhoGTPases independently of the activation status [61,78,82,95,191]. We observed that cIAP1 stabilized the interaction of cdc42 with its regulator RhoGDIα, thereby controlling cdc42 homeostasis. Similarly, to RhoGDIα downregulation, cIAP1 depletion sped up the activating cycle and turnover of cdc42, resulting in enhanced filopodia formation in unstimulated cells. On the other hand, cIAP1 functions as an intermediate between membrane receptor and cdc42 activation; therefore, downregulation of cIAP1 completely blocked TNF, EGF, and HRas-V12-mediated cdc42 activation and migration [61]. Thus, as observed in RIP-dependent signaling pathways, IAPs have the ability to negatively or positively regulate RhoGTPase activity and turnover in function of cellular environment and conditions. 

## 10. Conclusions

Numerous studies have demonstrated the critical function for IAPs in immune response. IAP-mediated ubiquitination constitutes the foundations of ubiquitin scaffold required for the assembly of multiprotein complexes that engage signaling from several receptors involved in immune response. IAP-mediated ubiquitination of RIPK1 and RIPK2 in receptor-associated complex has been well documented; however, other adapters could also be IAP ubiquitination targets. For example, cIAPs-mediated ubiquitination of Bcl10 has been shown to be critical for driving BCR-mediated NF-κB activation [91]. Besides receptor complex, IAPs have also the ability to regulate downstream effectors, acting at different levels of signaling, from signaling kinases to transcription factors and cytoskeleton regulators. IAPs participate in signal transduction leading to cell survival, differentiation, motility, migration, and cytokine production. Thanks to their ability to promote different types of ubiquitination on substrates, they are also able to block the signaling pathways either by promoting UPS-dependent degradation of key intermediates or by switching the signal from survival to cell death. Thus, IAPs are able to modulate the strength and kinetics of signaling and to modify the cell response to a specific signal in time and space. They constitute fast and efficient adjustment parameters allowing the adaption of cells to changing conditions. The identification of IAP partners involved in cell cycle regulation, DNA damage response, copper metabolism, or autophagy (Table 1) suggests that IAPs could have more general functions in adaptive response to cellular stress. 

Many of the IAP substrates remain to be identified, and the role and mechanisms of IAP-mediated ubiquitination are still misunderstood for many known partners. Determining the type and site of ubiquitination is of importance since it determines the cellular fate of proteins. Further works should be able to benefit from advances in understanding of the ubiquitin biology and from the development of new tools for investigating ubiquitination. 

The mechanisms of activation and regulation of IAPs are still an important issue. IAP E3-ubiquitine ligases are actives in a dimeric form. Since IAPs are able to form heteromeric complex [192], it could be interesting to determine whether homodimers and heterodimers could have different functions. In most case, when investigated, the interplay between TRAF2 and cIAP1 was observed. TRAF2 appeared to be a very important regulator of cIAPs [55,56,63]. It has been observed expressed both in the cytoplasmic and nuclear fraction. Whether TRAF2 is dispensable or absolutely required for the E3-ligase activity of IAPs is still an important question to address. Some post-translational modifications of IAPs have been described, but the number of studies remains limited. Upon CD40, BAFF, or TLR4 stimulation, cIAP E3-ubiquitine ligases are activated by TRAF2 or TRAF6-mediated K63-linked ubiquitination [63,101,184]. Oxidation of the Cys308 residue in the BIR3 domain seems sufficient to trigger the dimerization, activation, and auto-ubiquitination of cIAP1 [193]. S-nitrosylation of XIAP at Cys213 within the BIR2 can modulate the anti-apoptosis function of XIAP, while S-nitrosylation of cIAP1 at Cys571 and 574 located in the RING can inhibit its E3-ubiquitine ligase activity [194]. The phosphorylation of XIAP at Ser87 or Ser40 by PKC (protein kinase C) or cdk1/Cyclin B1 complex, respectively, could modulate its stability and anti-apoptotic function [195,196]. The subcellular localization of IAPs seems also of importance. Most IAP substrates are cytoplasmic proteins; however, we have found that cIAP1 can be recruited to some promoters [105,189], and cIAP1/2 and XIAP are able to ubiquitinate transcription factors and modulate their binding to chromatin [104,106,189]. While cIAPs are mostly located in the cytoplasm in immune cells, including lymphocytes and macrophages, we and other detected them in the nucleus of some cancer cells and undifferentiated cells (e.g., hematopoietic stem cells), and cIAP1 undergoes nuclear cytoplasmic translocation along a differentiation program [197,198,199]. 

IAPs appeared as pleiotropic proteins with cell type-specific functions. Accordingly, they have been involved in diverse human pathologies, including immune deficiencies, cancers, and inflammatory disorders [200], and IAP antagonists have entered into clinical trials for diverse clinical applications. It will be important to decipher further the specific functions and regulation mechanisms of IAPs in immune or non-immune cell subsets and to determine the impact of IAP antagonists on specific cell populations. 

## Figures and Tables

**Figure 1 cells-09-01118-f001:**
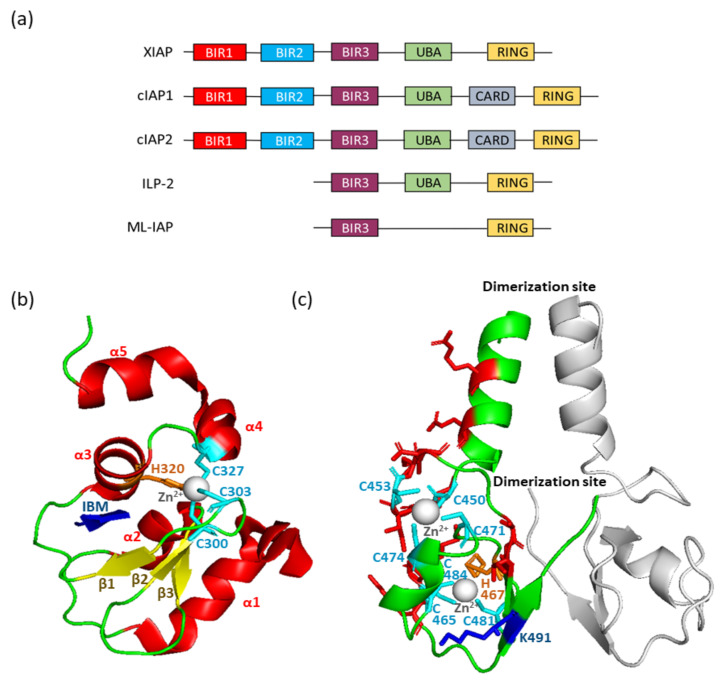
(**a**) Structure of Inhibitor of Apoptosis (IAP) E3-ubiquitine ligases. BIR: Baculovirus IAP repeat; UBA: ubiquitin associated; CARD: caspase activation and recruitment domain; RING: really interesting gene. (**b**) Ribbon diagram of the cIAP1 BIR3 domain in contact with the IAP binding motif (IBM) of caspase 9. The sheets (β) are shown as yellow ribbons and helixes (α) as red ribbons. β-strands and the third α-helix form a deep hydrophobic groove stabilized by zinc atom (grey spheres) that is coordinated by three conserved Cysteine (cyan) and one Histidine (orange) residues. The IBM is represented in blue. (**c**) Ribbon diagram of an X-linked IAP (XIAP)-RING domain homodimer. The zinc ions are shown as grey spheres, the cysteine residues interacting with zinc as cyan sticks, the histidine as orange sticks, the ubiquitin-interacting site as blue sticks, and the E2-interacting sites as red sticks.

**Figure 2 cells-09-01118-f002:**
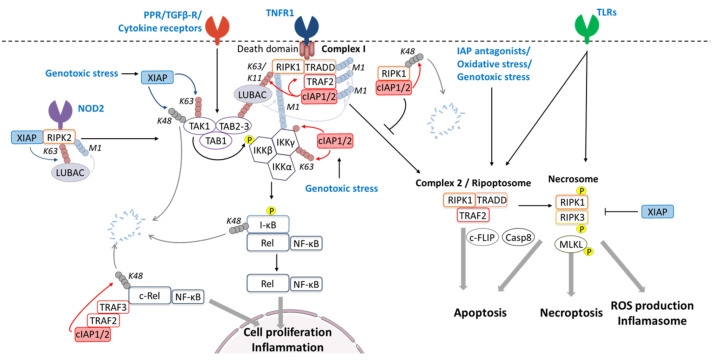
Regulation of the TNFR and canonical NF-κB signaling pathways by IAPs. The activation of the tumor necrosis factor receptor 1 (TNFR1) induces the formation of the complex 1 composed of TNFR-associated death domain (TRADD), receptor-interacting kinase 1 (RIPK1), TNFR-associated factor 2 (TRAF2), and cellular inhibitors of apoptosis (cIAP1/2). The latter promote the ubiquitination of RIPK1 (K63 and K11-linked), leading to the recruitment of the linear ubiquitin chain assembly complex (LUBAC). In turn, LUBAC induces the conjugation of linear M1-linked polyubiquitin chains to complex 1 components. The resulting ubiquitin scaffold allows the recruitment of TAK1 (tumor growth factor-β-activated kinase 1)/TAB1/TAB2/3 (transforming growth factor-activated kinase1-binding protein 1, 2, and 3) complex and IKK (Inhibitor of κB kinase) complex composed of IKKα/IKKβ/IKKγ. TAK1 catalyses the phosphorylation of IKKβ and the activation of IKK. This leads to the phosphorylation of the Inhibitor of κB (I-κB), its K48-linked ubiquitination, and proteasomal degradation, resulting in the release and nuclear translocation of NF-κB subunits. Stimulation of PRRs (pattern-recognition receptors), TGFβ-R (Transforming Growth Factor-beta receptor), cytokine receptors, NOD2 (nucleotide-binding oligomerization domain), or genotoxic stress can engage the canonical NF-κB pathways by stimulating TAK1/TAB1/TAB 2/3 complex. XIAP (X-linked inhibitor of apoptosis protein) can positively regulate TAK1 activation through K63-linked ubiquitination or negatively through K48-linked ubiquitination and proteasomal degradation. In the case of NOD2 stimulation, XIAP promotes the K63-linked ubiquitination of RIPK2 that results in the recruitment of LUBAC and TAK1 activation. cIAPs can control IKKγ by promoting mono- or K63-linked ubiquitination, while the cIAPs/TRAFs E3-ubiquitin ligase complex can regulate c-Rel stability. A complex 2 derived from TNFR1-associated complex 1 or Ripoptosome that contain RIPK1 and RIPK3 can be formed in response to depletion of cIAPs by IAP antagonists, oxidative or genotoxic stresses, or TLRs (toll-like receptors) stimulation. Theses complexes can lead to apoptosis, necroptosis, or ROS production and inflammasome activation, depending on presence of the regulatory or effector proteins c-FLIP (cellular FLICE-like inhibitory protein), caspase 8 (Casp8), and/or MLKL. cIAPs block complex 2/ripoptosme assembly by inhibiting RIPK1 kinase activity and/or by mediating its K48-linked ubiquitination and degradation. Necrosome is negatively controlled by XIAP.

**Figure 3 cells-09-01118-f003:**
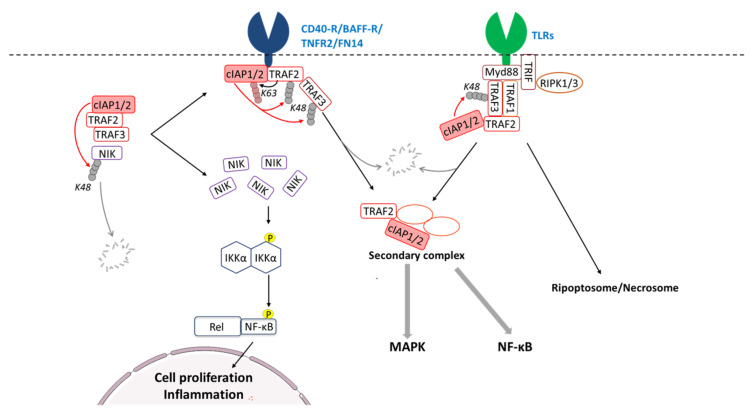
Regulation of signaling pathways by cIAP-mediated degradation of TRAF2 and/or TRAF3. In standing state, the cIAPs/TRAF2/3 E3-ubiquitin ligase complex controls the cellular content NIK (NF-κB-inducing kinase) by mediating K43-linked ubiquitination and proteasomal degradation. Stimulation of CD40-R (cluster of differentiation 40 receptor), BAFF-R (B-cell activating factor receptor), TNFR2 (tumor necrosis factor Receptor 2), FN14 (Fibroblast growth factor-inducible 14), or TLRs (toll-like receptors) promotes the recruitment of cIAPs/TRAFs complex to the receptor leading to the accumulation of NIK, phosphorylation of IKKα dimer, and activation of the non-canonical NF-κB pathway. In receptor-associated complex, TRAF2 can stimulate cIAPs via K63-linked ubiquitination. In turn, cIAPs catalyse the K48-linked polyubiquitination of TRAF3 and, in a lower level, TRAF2, causing their degradation and the formation of a secondary complex. Upon TLR stimulation, RIPK1/3 (receptor-interacting kinase 1/3) are also recruited to the receptor complex that can lead to the assembly of Ripoptosome and/or necrosome. Myd88: Myeloid differentiation primary response 88; TRIF: toll–interleukin 1 receptor domain-containing adaptor inducing IFN-β.

**Table 1 cells-09-01118-t001:** IAP ubiquitination substrates.

Cellular Process /Protein Family	Substrate or Partner	IAPs	Interacting Mode	Nature of the Ubiquitination/Consequences	Ref
Apoptosis	AIF	XIAP	BIR2	Degradative polyubiquitination & Nondegradative polyubiquitination at K255 interfering with AIF DNA binding.	[64,65]
	ARTS	XIAP	BIR3	Degradative polyubiquitination at K3, blocks XIAP inhibition	[66]
	Bcl2	XIAP	Indirect, via ARTS	Degradative ubiquitination favoring cell death	[46]
	Caspase-3 and -7	cIAP2	Direct	Mono-ubiquitination in in vitro assay	[16]
		cIAP1, XIAP	BIR2, IBM-dep.	K48-linked polyubiquitination inhibiting cell death	[67,68]
	Caspase-7	XIAP		Neddylation, inhibition of caspase activity	[21]
	Caspase-8	cIAP1, XIAP		Degradative ubiquitination of component of the Ripoptosome	[69]
	cIAP1/2	cIAP1		Degradative (auto)ubiquitination, required TRAF2	[70,71]
	FAF1	XIAP	BIR1-3	Degradative polyubiquitination, inhibits FAF1-mediated cell death	[72]
	FLIP_L_	cIAP1, XIAP		Degradative ubiquitination of component of the Ripoptosome. Ubiquitination of FLIP required caspase-8-mediated cleavage.	[69]
	Smac	cIAPs, XIAP	BIR2,3, IBM-dep.	Degradative ubiquitination, blocks XIAP inhibition and inhibits cell death	[73,74]
	XIAP	XIAP		Degradative autoubiquitination	[70]
Autophagy	p62	XIAP	Direct	Degradative ubiquitination	[75]
Copper homeostasis	CCS	XIAP	BIR3	Polyubiquitination at K241, enhances chaperone activity	[76]
	COMMD1	XIAP	BIR3	K48-linked ubiquitination, Increases intracellular copper in cultured cell	[77]
Cytoskeleton regulation	cdc42	XIAP	Direct	Degradative polyubiquitination at K166	[78]
		cIAP1	BIR2	Stabilization of RhoGDIα-cdc42 interaction. Regulation of the TNFα, EGR or Ras-V12-mediated activation	[61]
	Profilin 2	cIAP1	Direct	Degradative polyubiquitination	[79]
	Rac1	XIAP, cIAPs	BIR1-2	Degradative polyubiquitination at K147	[80]
				Upregulation and activation of Rac1	[81]
	RhoA	XIAP	Direct	Regulation of protease-activated receptor-mediated RhoA activation	[82]
DNA damage response/ cell cycle regulation	Chk1	XIAP	BIR3, IBM-dep.	Positive or negative modulation of the stability, regulated by XAF1	[32,83]
	MRE11	cIAP2	BIRs	Direct or indirect degradative polyubiquitination	[84]
	p21	cIAP1		Direct or indirect modulation of the neddylation leading to degradation	[85]
	Survivin	XIAP	Direct	Degradative ubiquitination, regulated by XAF1	[86]
Inflammation NF-κB	Caspase-1	cIAP1/2	BIR1	K63-linked activating ubiquitination	[87]
	SOCS1	XIAP	BIR1	K63-linked ubiquitination leading to stabilization	[88]
	IKKε	cIAPs /TRAF2		K63-linked ubiquitination at K30, K401, essential of kinase activity.	[44]
	IKKγ	cIAP1	BIR2-3	K6-polyubiquitination, mono-ubiquitination at K285 and K63-linked polyubiquitination at K277 and K309 leading to NF-kB activation.	[18,19,89]
	NIK	cIAPs	BIR2 and indirect (TRAF2/3)	K48-linked degradative ubiquitination	[31]
RIPK family	RIPK1	cIAPs	Direct or indirect	K11 and K63-linked ubiquitination leading to LUBAC recruitment and NF-κB activation. Degradative K48-linked polyubiquitination.	[12,13,69,90]
	RIPK2	XIAP	BIR2	K63-linked ubiquitination leading to LUBAC and TAK1/TAB1/TAB2 complex recruitment and NF-κB activation in NOD signaling	[20]
	RIPK2/3	cIAPs, XIAP		Polyubiquitination (diverse ubiquitin chains) in vitro assay	[14]
	RIPK4	cIAP1	Direct	Polyubiquitination (diverse chains) at K51 and K145, in vitro assay.	[14]
Signal transduction	ACs	XIAP	Indirect via TRIP-Br1	K27-linked polyubiquitination at K1047 (AC1) leading to enhanced AC1 endocytosis and degradation	[15]
	Bcl10	cIAP1/2		K63-linked ubiquitination leading to recruitment of LUBAC and IKK complex and NF-kB activation in BCR signaling	[91]
	MEKK2/3	cIAP1, XIAP	BIR1-2	K63-linked ubiquitination that blocks the MEKK2/3-MEK5-ERK5 cascade	[92]
	MEKK2	XIAP	?	K48- and K63-linked ubiquitination, regulation of the bi-phasic NF-κB activation	[93]
	PTEN	XIAP	Direct	Degradative polyubiquitination	[94]
	RAF1	XIAP, cIAPs	BIR1-2	Degradative polyubiquitination	[95]
	TAK1	XIAP	Direct or indirect via TAB1	K63-linked ubiquitination that results in kinase activation/Degradative K48-linked ubiquitination.	[96,97]
TRAF family	TRAF2	cIAP1	BIR1	Degradative and non degradative polyubiquitination. Regulation of receptor complex mediated signaling.	[98,99]
	TRAF3	cIAPs	Indirect, via TRAF2	K48-linked degradative ubiquitination leading to non-canonical NF-kB activation.	[63,100,101]
	TRAF6	cIAPs	Indirect	K48-linked polyubiquitination. K63-linked ubiquitination. Regulation of receptor complex- mediated signaling.	[102]
Transcriptional program	CHOP	cIAP1	Direct	Degradative ubiquitination, prevents ER-stress mediated apoptosis.	[103]
	CREB	cIAPs	Indirect, via TRAF2/3	K48-linked degradative ubiquitination	[45]
	c-Rel	cIAPs	Indirect, via TRAF2/3	K48-linked ubiquitination	[43]
	E2F1	cIAP1	BIR3	K63-linked ubiquitination at K161/164. Stabilization in S phase of cell cycle or upon genotoxic stress. Binding to chromatin	[104,105]
	Groucho	XIAP	Direct	Monoubiquitination leading to a decrease of the affinity of Groucho for transcription factor	[17]
	HIF1α	XIAP		K63-linked ubiquitination. Stabilization, nuclear translocation. Binding to chromatin	[106]
	IRF1	cIAP2	Direct	K63-linked ubiquitination leading to transcription factor activation. Regulated by S1P.	[107]
	IRF5	cIAPs	Indirect, via TRAF2/3	K48-linked ubiquitination	[43]
	Mad1	cIAP1	BIR1-3	Degradative ubiquitination. Activation of c-Myc	[108]

ACs (Adenylyl cyclased): enzyme generating cAMP; AIF (Apoptosis-inducing factor): regulator of chromatin condensation; ARTS: septin-like mitochondrial protein promoting apoptosis by antagonizing XIAP; Bcl2 (B-cell lymphoma 2): inhibitor of the intrinsic pathway of apoptosis; Bcl10 (B-cell lymphoma 10): adaptor protein in B cell receptor (BCR)-associated complex; BIR (baculovirus IAP repeat); CCS (copper chaperone for superoxide dismutase): responsible for delivery copper to superoxide dismutase; cdc42 (Cell Division Cycle 42): Small GTPase from Rho family regulator of actin cytoskeleton; Chk1 (Checkpoint kinase 1): Serine/Threonine protein kinase that controls the G2/M phase transition in response to DNA damage; CHOP (C/EBP homogous protein): transcription factor activated in response to endoplasmic reticulum stress; cIAP1 (cellular IAP); COMMD1 (Copper metabolism MURR1 domain protein 1): scaffold protein involved in copper homeostasis; CREB (C-AMP Response Element-binding protein): transcription factor involved in immune response; c-Rel: NF-κB subunit; E2F1 (E2 promoter binding factor 1): transcription factor involved in G1-S cell cycle phase transition and in DNA damage response; FAF1: Fas-associated Factor 1, enhances Fas-induced apoptosis; FLIP (FADD-like IL-1β-converting enzyme-inhibitory protein): major antiapoptotic protein; Groucho: repressing cofactor of the CTF/LEF (T-cell factor/lymphoid enhancer factor-1) transcription factor; HIF1α (Hypoxia-inducible factor-1α); IAP (Inhibitor of Apoptosis); IBM (IAP binding motif); IKK (I-κB kinase); IRF1&5 (Interferon (IFN) regulator factor 1&5): transcription factors involved in the induction of IFN, some inflammatory cytokines and genes of inflammation and immune response; Mad1 (Mitotic arrest deficient 1): spindle assembly checkpoint (SAC) regulator, repressor of c-myc; MEKK2/3 (Mitogen-Activated protein kinase 2/3): serine/threonine protein kinase involved in MAP kinase signaling pathway; MRE11 (Meiotic Recombination 11 Homolog): nuclease involved in the homologous recombination (HR) pathway of double strand break repair; NIK (NF-κB-inducing kinase); p21 (also named CIP: CDK-interacting protein 1): cyclin-dependent kinase inhibitor (CKI) that controls the S phase of cell cycle; p62: receptor for protein destined to be degraded by autophagy; Profilin 2: actin cytoskeletal regulator, mediator of synapse architecture; PTEN (phosphatase and tension homolog): regulator of the phosphatidylinositol 3-kinases (PI-3K)/Akt pathway; Rac1: Small GTPase from Rho family regulator of actin cytoskeleton; RAF1: serine/threonine-protein kinase (MAP3K) involved in Ras-RAF-ERK-MEK cascade; RIPK (receptor-interacting kinase); RhoA: Small GTPase from Rho family regulator of actin cytoskeleton; S1P (sphingosine-1-phosphate): bioactive sphingolipid; Smac (second mitochondria-derived activator of caspase): IAP inhibitor; SOCS1 (suppressor of cytokine signaling 1): inhibitor of IFNs and some cytokine signaling pathways; TAK1 (tumor growth factor-β-activated kinase 1); TRAF (tumor necrosis factor Receptor (TNFR)-associated factor); XIAP (X-linked IAP).

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
