# Peer review of "IAP-Mediated Protein Ubiquitination in Regulating Cell Signaling"

_cells, 2020, doi:10.3390/cells9051118_

Round 1

Reviewer 1 Report

The manuscript presents a comprehensive review of the role and mechanism of IAP-mediated ubiquitination in regulating intracellular signaling. It is well studied and readable. This reviewer has a few minor comments.

1. Fig.1(b)(c): The font size is small.

2. There are several typo errors. Please confirm the following:

P113: E3’ --> E3 (?)

P174: Fibroblasts --> fibroblasts

P321: N-termi --> N-terminus (?) 

P328: nuclear --> nuclear factor

P445: K377 residues --> K377

P555: 1a)) --> 1a)

Author Response

We thank the referee for the positive analysis of our manuscript.

  1. We increased the police size for Figure 1 and also for Figure 2 and 3
  2. 2. We corrected the typo errors

Reviewer 2 Report

The manuscript, titled “IAP-mediated protein ubiquitination in regulating cell signaling”, is good topic regarding IAP-mediated ubiquitination and would guide researchers, in the field of IAP, a new ideas to deeply study as well as help the researchers, in the field of regulation cell signaling, to extent his/her research. Even though mentioned above, I still have a couple of concerns below:

  1. I checked the plagiarism and found it’s 51%, please make change the language to be satisfy with the requirement of the journal.
  2. Laurence Dubrez published many papers regarding IAP, please state their works in the manuscript.
  3. Line 24 to 25, “IAPs (Inhibitors of Apoptosis) constitute a family of conserved proteins found in a variety of 24 organisms including yeast, virus, nematodes, insect, fishes and mammals” need to be provided references for different species. Same concern in line 27 “Mammal IAPs share this anti-apoptotic property”.
  4. Line 28: please define XIAP.
  5. Line 54 to 55: “a few is known about their function in other signaling pathways”, please give some other signaling pathways’ name.
  6. Table 1: remove some parts from column of cellular process/protein family, if you didn't discuss them in your subtitles such as Autophage, Copper homeostasis, DNA damage response/cell cycle regulation and signal transduction.
  7. Searching the IAP in latest 5 years, I got 933 articles published, but the manuscript just cited 28. To consider the timeline and current related achievements, please read more latest articles.

Author Response

Comments and Suggestions for Authors

The manuscript, titled “IAP-mediated protein ubiquitination in regulating cell signaling”, is good topic regarding IAP-mediated ubiquitination and would guide researchers, in the field of IAP, a new ideas to deeply study as well as help the researchers, in the field of regulation cell signaling, to extent his/her research. Even though mentioned above, I still have a couple of concerns below:

  1. I checked the plagiarism and found it’s 51%, please make change the language to be satisfy with the requirement of the journal.

We used the MDPI English edition service to check the plagiarism (please see the attachment). Indeed, the result is 50%. However, it includes the references (which represent about half of the article) and the development of abbreviation and scientific term as shown below.

We carefully checked the document and we only detected two complete sentences under lighted which are:

  • They are both regulated by phosphorylation and ubiquitination processes
  • TAK1 is a mitogen-activated protein kinase kinase kinase (MAP3K)

We removed the first sentence.

  1. Laurence Dubrez published many papers regarding IAP, please state their works in the manuscript.

We stated our work related to the cIAP1-mediated degradation of TRAF2 along macrophage differentiation in lines 516-518 and we indicated our results on the nuclear expression of cIAP1 and its recruitment to gene promoters in lines 659-664. Our studies demonstrating (i) the ability of cIAP1 to regulate E2F1 transcription factors is detailed in lines 576-583 and (ii) the ability of cIAP1 to regulated cdc42 in lines 610-615.

  1. Line 24 to 25, “IAPs (Inhibitors of Apoptosis) constitute a family of conserved proteins found in a variety of organisms including yeast, virus, nematodes, insect, fishes and mammals” need to be provided references for different species. Same concern in line 27 “Mammal IAPs share this anti-apoptotic property”.

We provided some references as requested

  1. Line 28: please define XIAP.

We developped XIAP

  1. Line 54 to 55: “a few is known about their function in other signaling pathways”, please give some other signaling pathways’ name.

We added “(e.g. autophagy, DNA damage response, transcriptional regulation…)”

  1. Table 1: remove some parts from column of cellular process/protein family, if you didn't discuss them in your subtitles such as Autophage, Copper homeostasis, DNA damage response/cell cycle regulation and signal transduction.

Although less documented, I find important to present these IAP ubiquitination substrates and to indicate the involvement of IAPs in these cellular processes because they are rarely presented in the reviews on IAPs. This is one of the originality of this review.

  1. Searching the IAP in latest 5 years, I got 933 articles published, but the manuscript just cited 28. To consider the timeline and current related achievements, please read more latest articles.

Most of publications on IAPs published during the last 5 years are focused on IAP antagonists or IAP inhibitors and their efficiency in disease treatment. Moreover, only 5 out of the 8 mammals IAP are E3-ubiquitine ligases. The publications related to IAP ubiquitination substrates are a minor part of the publications on IAPs.

This review is focused on the IAP ubiquitination substrates. Therefore, I selected the articles that identified ubiquitination substrates, characterized the mechanisms and the type of ubiquitination and that demonstrated the role of IAP-mediated ubiquitination in signaling pathways. I have chosen to refer the first article that identified the IAP substrate instead of the last one. I changed some publications in order to indicated more recent articles and added new publications.

Considering the high number of references in the article, it is not possible to cited all of them. I apologized the authors that made contributions to the field but have not been cited due to space limitations in the Acknowledgments section.
